# Excessive Invariance Causes Adversarial Vulnerability

**Jörn-Henrik Jacobsen** [1]*, **Jens Behrmann**[1,2], **Richard Zemel**[1], **Matthias Bethge**[3]
[1]Vector Institute and University of Toronto
[2]University of Bremen, Center for Industrial Mathematics
[3]University of Tübingen
*j.jacobsen@vectorinstitute.ai

## Abstract

Despite their impressive performance, deep neural networks exhibit striking failures on out-of-distribution inputs. One core idea of adversarial example research is to reveal neural network errors under such distribution shifts. We decompose these errors into two complementary sources: sensitivity and invariance. We show deep networks are not only *too sensitive* to task-irrelevant changes of their input, as is well-known from $\epsilon$-adversarial examples, but are also *too invariant* to a wide range of task-relevant changes, thus making vast regions in input space vulnerable to adversarial attacks. We show such excessive invariance occurs across various tasks and architecture types. On MNIST and ImageNet one can manipulate the class-specific content of almost any image without changing the hidden activations. We identify an insufficiency of the standard cross-entropy loss as a reason for these failures. Further, we extend this objective based on an information-theoretic analysis so it encourages the model to consider all task-dependent features in its decision. This provides the first approach tailored explicitly to overcome excessive invariance and resulting vulnerabilities.

## 1 Introduction

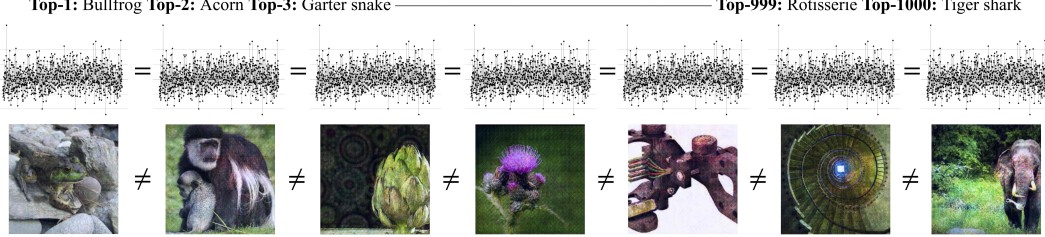

Figure 1: All images shown cause a competitive ImageNet-trained network to output the *exact same* probabilities over all 1000 classes (logits shown above each image). The leftmost image is from the ImageNet validation set; all other images are constructed such that they match the non-class related information of images taken from other classes (for details see section 2.1). The excessive invariance revealed by this set of adversarial examples demonstrates that the logits contain only a small fraction of the information perceptually relevant to humans for discrimination between the classes.

Adversarial vulnerability is one of the most iconic failure cases of modern machine learning models (Szegedy et al., 2013) and a prime example of their weakness in out-of-distribution generalization. It is particularly striking that under i.i.d. settings deep networks show superhuman performance on many tasks (LeCun et al., 2015), while tiny targeted shifts of the input distribution can cause them to make unintuitive mistakes. The reason for these failures and how they may be avoided or at least mitigated is an active research area (Schmidt et al., 2018; Gilmer et al., 2018b; Bubeck et al., 2018).

So far, the study of adversarial examples has mostly been concerned with the setting of small perturbation, or $\epsilon$-adversaries (Goodfellow et al., 2015; Madry et al., 2017; Raghunathan et al., 2018).

Perturbation-based adversarial examples are appealing because they allow to quantitatively measure notions of adversarial robustness (Brendel et al., 2018). However, recent work argued that the perturbation-based approach is unrealistically restrictive and called for the need of generalizing the concept of adversarial examples to the unrestricted case, including any input crafted to be misinterpreted by the learned model (Song et al., 2018; Brown et al., 2018). Yet, settings beyond $\epsilon$-robustness are hard to formalize (Gilmer et al., 2018a).

We argue here for an alternative, complementary viewpoint on the problem of adversarial examples. Instead of focusing on transformations erroneously crossing the decision-boundary of classifiers, we focus on excessive invariance as a major cause for adversarial vulnerability. To this end, we introduce the concept of invariance-based adversarial examples and show that class-specific content of almost any input can be changed arbitrarily without changing activations of the network, as illustrated in figure 1 for ImageNet. This viewpoint opens up new directions to analyze and control crucial aspects underlying vulnerability to unrestricted adversarial examples.

The invariance perspective suggests that adversarial vulnerability is a consequence of narrow learning, yielding classifiers that rely only on few highly predictive features in their decisions. This has also been supported by the observation that deep networks strongly rely on spectral statistical regularities (Jo & Bengio, 2017), or stationary statistics (Gatys et al., 2017) to make their decisions, rather than more abstract features like shape and appearance. We hypothesize that a major reason for this excessive invariance can be understood from an information-theoretic viewpoint of cross-entropy, which maximizes a bound on the mutual information between labels and representation, giving no incentive to explain all class-dependent aspects of the input. This may be desirable in some cases, but to achieve truly general understanding of a scene or an object, machine learning models have to learn to successfully separate essence from nuisance and subsequently generalize even under shifted input distributions.

Our contributions:

- We identify excessive invariance underlying striking failures in deep networks and formalize the connection to adversarial examples.

- We show invariance-based adversarial examples can be observed across various tasks and types of deep network architectures.

- We propose an invertible network architecture that gives explicit access to its decision space, enabling class-specific manipulations to images while leaving all dimensions of the representation seen by the final classifier invariant.

- From an information-theoretic viewpoint, we identify the cross-entropy objective as a major reason for the observed failures. Leveraging invertible networks, we propose an alternative objective that provably reduces excessive invariance and works well in practice.

## 2 TWO COMPLEMENTARY APPROACHES TO ADVERSARIAL EXAMPLES

In this section, we define pre-images and establish a link to adversarial examples.

**Definition 1** (Pre-images / Invariance). *Let $F : \mathbb{R}^d \to \mathbb{R}^C$ be a neural network, $F = f_L \circ \cdots \circ f_1$ with layers $f_i$ and let $F_i$ denote the network up to layer $i$. Further, let $D : \mathbb{R}^d \to \{1, \ldots, C\}$ be a classifier with $D = \arg\max_{k=1,\ldots,C} softmax(F(x))_k$. Then, for input $x \in \mathbb{R}^d$, we define the following pre-images*

*(i)  i-th Layer pre-image: $\{x^* \in \mathbb{R}^d \mid F_i(x^*) = F_i(x)\}$*

*(ii)  Logit pre-image: $\{x^* \in \mathbb{R}^d \mid F(x^*) = F(x)\}$*

*(iii)  Argmax pre-image: $\{x^* \in \mathbb{R}^d \mid D(x^*) = D(x)\}$,*

*where (i) $\subset$ (ii) $\subset$ (iii) by the compositional nature of $D$.*
*Moreover, the (sub-)network is **invariant** to perturbations $\Delta x$ which satisfy $x^* = x + \Delta x$.*

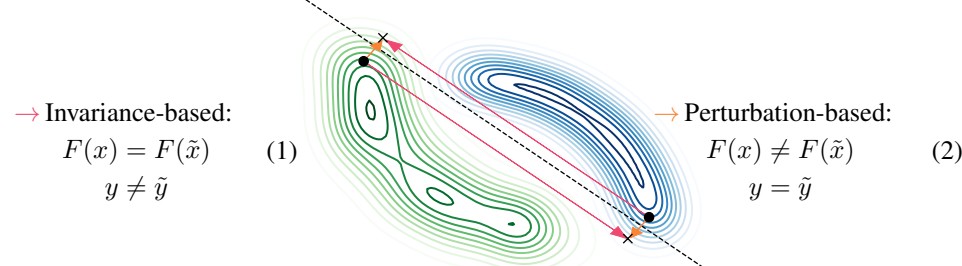

$\rightarrow$ Invariance-based:
$$F(x) = F(\tilde{x}) \qquad (1)$$
$$y \neq \tilde{y}$$

$\rightarrow$ Perturbation-based:
$$F(x) \neq F(\tilde{x}) \qquad (2)$$
$$y = \tilde{y}$$

Figure 2: Connection between (1) invariance-based (long pink arrow) and (2) perturbation-based adversarial examples (short orange arrow). Class distributions are shown in green and blue; dashed line is the decision-boundary of a classifier. All adversarial examples can be reached either by crossing the decision-boundary of the classifier via perturbations, or by moving within the pre-image of the classifier to mis-classified regions. The two viewpoints are *complementary* to one another and highlight that adversarial vulnerability is not only caused by excessive *sensitivity* to semantically meaningless perturbations, but also by excessive *insensitivity* to semantically meaningful transformations.

Non-trivial pre-images (pre-images containing more elements than input $x$) after the $i$-th layer occur if the chain $f_i \circ \cdots \circ f_1$ is not injective, for instance due to subsampling or non-injective activation functions like ReLU (Behrmann et al., 2018a). This accumulated invariance can become problematic if not controlled properly, as we will show in the following.

We define perturbation-based adversarial examples by introducing the notion of an oracle (e.g., a human decision-maker or the unknown input-output function considered in learning theory):

**Definition 2** (Perturbation-based Adversarial Examples). *A **Perturbation-based adversarial example** $x^* \in \mathbb{R}^d$ of $x \in \mathbb{R}^d$ fulfills:*

(i) *Perturbation of decision: $D(x^*) \neq o(x^*)$ and $D(x) \neq D(x^*)$, where $D : \mathbb{R}^d \to \{1, \ldots, C\}$ is the **classifier** and $o : \mathbb{R}^d \to \{1, \ldots, C\}$ is the **oracle**.*

(ii) *Created by adversary: $x^* \in \mathbb{R}^d$ is created by an algorithm $\mathcal{A} : \mathbb{R}^d \to \mathbb{R}^d$ with $x \mapsto x^*$.*

*Further, $\epsilon$-**bounded** adversarial ex. $x^*$ of $x$ fulfill $\|x - x^*\| < \epsilon$, $\|\cdot\|$ a norm on $\mathbb{R}^d$ and $\epsilon > 0$.*

Usually, such examples are constructed as $\epsilon$-bounded adversarial examples (Goodfellow et al., 2015). However, as our goal is to characterize general invariances of the network, we do not restrict ourselves to bounded perturbations.

**Definition 3** (Invariance-based Adversarial Examples). *Let $G$ denote the $i$-th layer, logits or the classifier (Definition 1) and let $x^* \neq x$ be in the $G$ pre-image of $x$ and and $o$ an oracle (Definition 2). Then, an **invariance-based adversarial example** fulfills $o(x) \neq o(x^*)$, while $G(x) = G(x^*)$ (and hence $D(x) = D(x^*)$).*

Intuitively, adversarial perturbations cause the output of the classifier to change while the oracle would still consider the new input $x^*$ as being from the original class. Hence in the context of $\epsilon$-bounded perturbations, the classifier is *too sensitive to task-irrelevant changes*. On the other hand, movements in the pre-image leave the classifier invariant. If those movements induce a change in class as judged by the oracle, we call these invariance-based adversarial examples. In this case, however, the classifier is *too insensitive to task-relevant changes*. In conclusion, these two modes are complementary to each other, whereas both constitute failure modes of the learned classifier.

When not restricting to $\epsilon$-perturbations, perturbation-based and invariance-based adversarial examples yield the same input $x^*$ via

$$x^* = x_1 + \Delta x_1, \quad D(x^*) \neq D(x_1), \quad o(x^*) = o(x_1) \qquad (3)$$
$$x^* = x_2 + \Delta x_2, \quad D(x^*) = D(x_2), \quad o(x^*) \neq o(x_2), \qquad (4)$$

with different reference points $x_1$ and $x_2$, see Figure 2. Hence, the **key difference is the change of reference**, which allows us to approach these failure modes from different directions. To connect these failure modes with an intuitive understanding of variations in the data, we now introduce the notion of invariance to nuisance and semantic variations, see also (Achille & Soatto, 2018).

**Definition 4** (Semantic/ Nuisance perturbation of an input). *Let $o$ be an oracle (Definition 2) and $x \in \mathbb{R}^d$. Then, a perturbation $\Delta x$ of an input $x \in \mathbb{R}^d$ is called **semantic**, if $o(x) \neq o(x + \Delta x)$ and **nuisance** if $o(x) = o(x + \Delta x)$.*

For example, such a nuisance perturbation could be a translation or occlusion in image classification. Further in Appendix A, we discuss the synthetic example called *Adversarial Spheres* from (Gilmer et al., 2018b), where nuisance and semantics can be explicitly formalized as rotation and norm scaling.

## 2.1 USING BIJECTIVE NETWORKS TO ANALYZE EXCESSIVE INVARIANCE

As invariance-based adversarial examples manifest themselves in changes which do not affect the output of the network $F$, we need a generic approach that gives us access to the discarded nuisance variability. While feature nuisances are intractable to access for general architectures (see comment after Definition 1), invertible classifiers only remove nuisance variability in their final projection (Jacobsen et al., 2018). For $C < d$, we denote the classifier as $D : \mathbb{R}^d \to \{1, ..., C\}$. Our contributions in this section are: **(1)** Introduce an invertible architecture with a simplified readout structure, allowing to exactly visualize manipulations in the hidden-space, **(2)** Propose an analytic attack based on this architecture allowing to analyze its decision-making, **(3)** Reveal striking invariance-based vulnerability in competitive classifiers.

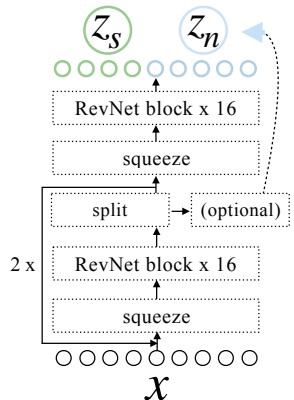

Figure 3: The fully invertible RevNet, a hybrid of Glow and iRevNet with simple readout structure. $z_s$ represents the logits and $z_n$ the nuisance.

**Bijective classifiers with simplified readout.** We build deep networks that give access to their decision space by removing the final linear mapping onto the class probes in invertible RevNet-classifiers and call these networks fully invertible RevNets. The fully invertible RevNet classifier can be written as $D_\theta = \arg\max_{k=1,...,C} softmax(F_\theta(x)_k)$, where $F_\theta$ represents the bijective network. We denote $z = F_\theta(x)$, $z_s = z_{1,...,C}$ as the logits (semantic variables) and $z_n = z_{C+1,...,d}$ as the nuisance variables ($z_n$ is not used for classification). In practice we choose the first C indices of the final $z$ tensor or apply a more sophiscticated DCT scheme (see appendix D) to set the subspace $z_s$, but other choices work as well. The architecture of the network is similar to iRevNets (Jacobsen et al., 2018) with some additional Glow components like actnorm (Kingma & Dhariwal, 2018), squeezing, dimension splitting and affine block structure (Dinh et al., 2017), see Figure 3 for a graphical description. As all components are common in the bijective network literature, we refer the reader to Appendix D for exact training and architecture details. Due to its simple readout structure, the resulting invertible network allows to qualitatively and quantitatively investigate the task-specific content in nuisance and logit variables. Despite this restriction, we achieve performance on par with commonly-used baselines on MNIST and ImageNet, see Table 1 and Appendix D.

| % Error | fi-RevNet48(Ours) | VGG19 | ResNet18 | ResNet50 | iRevNet300 |
|---|---|---|---|---|---|
| ILSVRC2012 Val Top1 | 29.50 | 28.70 | 30.43 | 24.70 | 26.70 |
| ILSVRC2012 Val Top5 | 11.30 | 9.90 | 10.80 | 7.89 | - |

Table 1: The table shows error rates on the ILSVRC-2012 validation set of our proposed fully invertible RevNet compared to a VGG (Simonyan & Zisserman, 2014) and two ResNet (He et al., 2016) variants, as well as an iRevNet (Jacobsen et al., 2018) with a non-invertible final projection onto the logits. Our proposed fully invertible RevNet performs roughly on par with others.

**Analytic attack.** To analyze the trained models, we can sample elements from the logit pre-image by computing $x_{met} = F^{-1}(z_s, \tilde{z}_n)$, where $z_s$ and $\tilde{z}_n$ are taken from two different inputs. We term this heuristic *metameric sampling*. The samples would be from the true data distribution if the subspaces would be factorized as $P(z_s, z_n) = P(z_s)P(z_n)$. Experimentally we find that logit metamers are revealing adversarial subspaces and are visually close to natural images on ImageNet.

**Clean Subspace**                    **Adversarial Subspace**

outer
inner

Logit Classifier        Nuisance Classifier        Logit Classifier        Nuisance Classifier

Figure 4: Left: Decision-boundaries in 2D subspace spanned by two random data points $x_1, x_2$. Right: Decision-boundaries in 2D subspace spanned by random datapoint $x$ and metamer $x_{met}$.

Thus, metameric sampling gives us an analytic tool to inspect dependencies between semantic and nuisance variables without the need for expensive and approximate optimization procedures.

**Attack on adversarial spheres.** First, we evaluate our analytic attack on the synthetic spheres dataset, where the task is to classify samples as belonging to one out of two spheres with different radii. We choose the sphere dimensionality to be $d = 100$ and the radii: $R_1 = 1$, $R_2 = 10$. By training a fully-connected fully invertible RevNet, we obtain 100% accuracy. After training we visualize the decision-boundaries of the original classifier $D$ and a posthoc trained classifier on $z_n$ (nuisance classifier), see Figure 4. We densely sample points in a 2D subspace, following Gilmer et al. (2018b), to visualize two cases: 1) the decision-boundary on a 2D plane spanned by two randomly chosen data points, 2) the decision-boundary spanned by metameric sample $x_{met}$ and reference point $x$. In the metameric sample subspace we identify excessive invariance of the classifier. Here, it is possible to move any point from the inner sphere to the outer sphere without changing the classifiers predictions. However, this is not possible for the classifier trained on $z_n$. Most notably, the visualized failure is not due to a lack of data seen during training, but rather due to excessive invariance of the original classifier $D$ on $z_s$. Thus, the nuisance classifier on $z_n$ does not exhibit the same adversarial vulnerability in its subspace.

**ResNet**                                      **fi-RevNet**

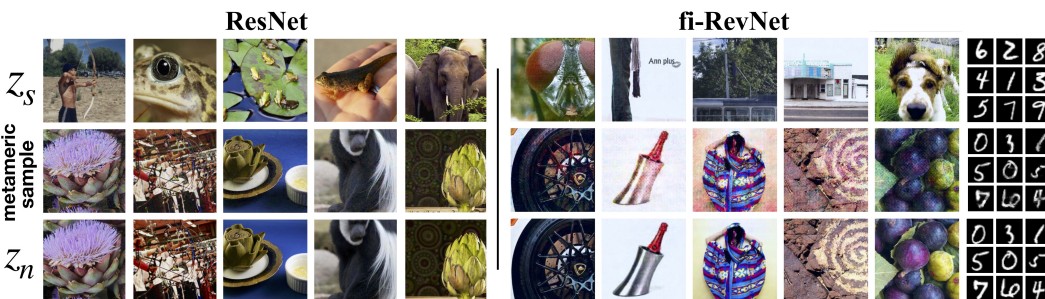

$z_s$

metameric sample

$z_n$

Figure 5: Each column shows three images belonging together. Top row are source images from which we sample the logits, middle row are logit metamers and bottom row images from which we sample the nuisances. Top row and middle row have the same (approximately for ResNets, exactly for fully invertible RevNets) logit activations. Thus, it is possible to change the image content completely without changing the 10- and 1000-dimensional logit vectors respectively. This highlights a striking failure of classifiers to capture all task-dependent variability.

**Attack on MNIST and ImageNet.** After validating its potential to uncover adversarial subspaces, we apply metameric sampling to fully invertible RevNets trained on MNIST and Imagenet, see Figure 5. The result is striking, as the nuisance variables $z_n$ are dominating the visual appearance of the logit metamers, making it possible to attach any semantic content to any logit activation pattern. Note that the entire 1000-dimensional feature vector containing probabilities over all ImageNet classses remains unchanged by any of the transformations we apply. To show our findings are not a particular property of bijective networks, we attack an ImageNet trained ResNet152 with a gradient-based version of our metameric attack, also known as feature adversaries (Sabour et al., 2016). The attack minimizes the mean squared error between a given set of logits from one image to another image (see appendix B for details). The attack shows the same failures for non-bijective models. This result highlights the general relevance of our finding and poses the question of the origin of this excessive invariance, which we will analyze in the following section.

# 3 OVERCOMING INSUFFICIENCY OF CROSSENTROPY-BASED INFORMATION-MAXIMIZATION

In this section we identify why the cross-entropy objective does not necessarily encourage to explain all task-dependent variations of the data and propose a way to fix this. As shown in figure 4, the nuisance classifier on $z_n$ uses task-relevant information not captured by the logit classifier $D_\theta$ on $z_s$ (evident by its superior performance in the adversarial subspace).

We leverage the simple readout-structure of our invertible network and turn this observation into a formal explanation framework using information theory: Let $(x, y) \sim \mathcal{D}$ with labels $y \in \{0, 1\}^C$. Then the goal of a classifier can be stated as maximizing the mutual information (Cover & Thomas, 2006) between semantic features $z_s$ (logits) extracted by network $F_\theta$ and labels $y$, denoted by $I(y; z_s)$.

**Adversarial distribution shift.** As the previously discussed failures required to modify input data from distribution $\mathcal{D}$, we introduce the concept of an *adversarial distribution shift* $\mathcal{D}_{Adv} \neq \mathcal{D}$ to formalize these modifications. Our first assumptions for $\mathcal{D}_{Adv}$ is $I_{\mathcal{D}_{Adv}}(z_n; y) \leq I_\mathcal{D}(z_n; y)$. Intuitively, the nuisance variables $z_n$ of our network do not become more informative about $y$. Thus, the distribution shift may reduce the predictiveness of features encoded in $z_s$, but does not introduce or increase the predictive value of variations captured in $z_n$. Second, we assume $I_{\mathcal{D}_{Adv}}(y; z_s|z_n) \leq I_{\mathcal{D}_{Adv}}(y; z_s)$, which corresponds to positive or zero interaction information, see e.g. (Ghassami & Kiyavash, 2017). While the information in $z_s$ and $z_n$ can be redundant in this assumption, synergetic effects where conditioning on $z_n$ increase the mutual information between $y$ and $z_s$ are excluded.

Bijective networks $F_\theta$ capture all variations by design which translates to information preservation $I(y; x) = I(y; F_\theta(x))$, see (Kraskov et al., 2004). Consider the reformulation

$$I(y; x) = I(y; F_\theta(x)) = I(y; z_s, z_n) = I(y; z_s) + I(y; z_n|z_s) = I(y; z_n) + I(y; z_s|z_n) \quad (5)$$

by the chain rule of mutual information (Cover & Thomas, 2006), where $I(y; z_n|z_s)$ denotes the conditional mutual information. Most strikingly, equation 5 offers two ways forward:

1. Direct increase of $I(y; z_s)$
2. Indirect increase of $I(y; z_s|z_n)$ via decreasing $I(y; z_n)$.

Usually in a classification task, only $I(y; z_s)$ is increased actively via training a classifier. While this approach is sufficient in most cases, expressed via high accuracies on training and test data, it may fail under $\mathcal{D}_{Adv}$. This highlights why cross-entropy training may not be sufficient to overcome excessive semantic invariance. However, by leveraging the bijection $F_\theta$ we can minimize the unused information $I(y; z_n)$ using the intuition of a nuisance classifier.

**Definition 5** (Independence cross-entropy loss). *Let $F_\theta : \mathbb{R}^d \rightarrow \mathbb{R}^d$ a bijective network with parameters $\theta \in \mathbb{R}^{p_1}$ and $\tilde{F}_\theta(x) = softmax(F_\theta(x)_{1,...,C})$. Furthermore, let $D_{\theta_{nc}} : \mathbb{R}^{d-C} \rightarrow [0, 1]^C$ be the nuisance classifier with $\theta_{nc} \in \mathbb{R}^{p_2}$. Then, the independence cross-entropy loss is defined as:*

$$\min_\theta \max_{\theta_{nc}} \mathcal{L}_{iCE}(\theta, \theta_{nc}) = \underbrace{\sum_{i=1}^C -y_i \log \tilde{F}_\theta^{z_s}(x)_i}_{=: \mathcal{L}_{sCE}(\theta)} + \underbrace{\sum_{i=1}^C y_i \log D_{\theta_{nc}}(F_\theta^{z_n}(x))_i}_{=: \mathcal{L}_{nCE}(\theta, \theta_{nc})}.$$

The underlying principles of the nuisance classification loss $\mathcal{L}_{nCE}$ can be understood using a variational lower bound on mutual information from Barber & Agakov (2003). In summary, the minimization is with respect to a lower bound on $I_\mathcal{D}(y; z_n)$, while the maximization aims to tighten the bound (see Lemma 10 in Appendix C). By using these results, we now state the main result under the assumed distribution shift and successful minimization (proof in Appendix C.1):

**Theorem 6** (Information $I_{\mathcal{D}_{Adv}}(y; z_s)$ maximal after distribution shift). *Let $\mathcal{D}_{Adv}$ denote the adversarial distribution and $\mathcal{D}$ the training distribution. Assume $I_\mathcal{D}(y; z_n) = 0$ by minimizing $\mathcal{L}_{iCE}$ and the distribution shift satisfies $I_{\mathcal{D}_{Adv}}(z_n; y) \leq I_\mathcal{D}(z_n; y)$ and $I_{\mathcal{D}_{Adv}}(y; z_s|z_n) \leq I_{\mathcal{D}_{Adv}}(y; z_s)$. Then,*

$$I_{\mathcal{D}_{Adv}}(y; z_s) = I_\mathcal{D}(y; x).$$

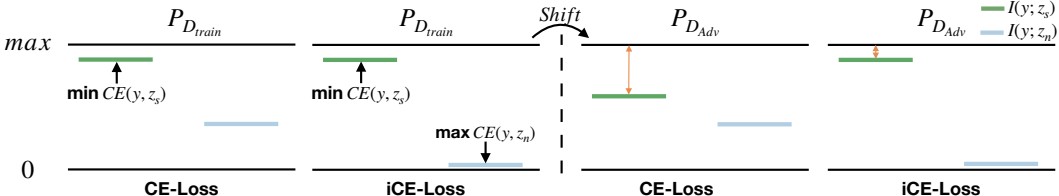

Figure 6: Left: Mutual information under distribution $\mathcal{D}_{train}$, Right: Effect of distributional shift to $\mathcal{D}_{Adv}$. Each case under training with cross-entropy (CE) and independence cross-entropy (iCE). Under distribution $\mathcal{D}$, the iCE-loss minimizes $I(y; z_n)$ (Lemma 10, Appendix C), but has no effect as the CE-loss already maximizes $I(y; z_s)$. However under the shift to $\mathcal{D}_{Adv}$, the information $I(y; z_s)$ decreases when training only under the CE-loss (orange arrow), while the iCE-loss induces $I(y; z_n) = 0$ and thus leaves $I(y; z_s)$ unchanged (Theorem 6).

Thus, incorporating the nuisance classifier allows for the discussed indirect increase of $I_{\mathcal{D}_{Adv}}(y; z_s)$ under an adversarial distribution shift, visualized in Figure 6.

To aid stability and further encourage factorization of $z_s$ and $z_n$ in practice, we add a maximum likelihood term to our independence cross-entropy objective as

$$\min_\theta \max_{\theta_{nc}} \mathcal{L}(\theta, \theta_{nc}) = \mathcal{L}_{iCE}(\theta, \theta_{nc}) - \underbrace{\sum_{k=1}^{d-C} \log\left(p_k(F_\theta^{z_n}(x)_k)|\det(J_\theta^x)|\right)}_{=:\mathcal{L}_{MLE_n}(\theta)}, \qquad (6)$$

where $\det(J_\theta^x)$ denotes the determinant of the Jacobian of $F_\theta(x)$ and $p_k \sim \mathcal{N}(\beta_k, \gamma_k)$ with $\beta_k, \gamma_k$ learned parameter. The log-determinant can be computed exactly in our model with negligible additional cost. Note, that optimizing $\mathcal{L}_{MLE_n}$ on the nuisance variables together with $\mathcal{L}_{sCE}$ amounts to maximum-likelihood under a factorial prior (see Lemma 11 in Appendix C).

Just as in GANs the quality of the result relies on a tight bound provided by the nuisance classifier and convergence of the MLE term. Thus, it is important to analyze the success of the objective after training. We do this by applying our metameric sampling attack, but there are also other ways like evaluating a more powerful nuisance classifier after training.

## 4 APPLYING INDEPENDENCE CROSS-ENTROPY

In this section, we show that our proposed independence cross-entropy loss is effective in reducing invariance-based vulnerability in practice by comparing it to vanilla cross-entropy training in four aspects: **(1)** error on train and test set, **(2)** effect under distribution shift, perturbing nuisances via metameric sampling, **(3)** evaluate accuracy of a classifier on the nuisance variables to quantify the class-specific information in them and **(4)** on our newly introduced shiftMNIST, an augmented version of MNIST to benchmark adversarial distribution shifts according to Theorem 6.

For all experiments we use the same network architecture and settings, the only difference being the two additional loss terms as explained in Definition 5 and equation 6. In terms of test error of the logit classifier, both losses perform approximately on par, whereas the gap between train and test error vanishes for our proposed loss function, indicating less overfitting. For classification errors see Table 2 in appendix D.

**Robustness under metameric sampling attack.** To analyze if our proposed loss indeed leads to independence between $z_n$ and labels $y$, we attack it with our metameric sampling procedure. As we are only looking on data samples and not on samples from the model (factorized gaussian on nuisances), this attack should reveal if the network learned to trick the objective. In Figure 7 we show interpolations between original images and logit metamers in CE- and iCE-trained fully invertible RevNets. In particular, we are holding the activations $z_s$ constant, while linearly interpolating nuisances $z_n$ down the column. The CE-trained network allows us to transform any image into any class without changing the logits. However, when training with our proposed iCE, the picture changes fundamentally and interpolations in the pre-image only change the style of a digit, but not its semantic content. This shows our loss has the ability to overcome excessive task-related invariance and encourages the model to explain and separate all task-related variability of the input from the nuisances of the task.

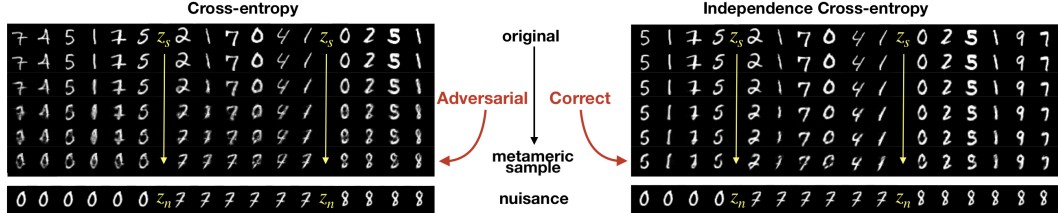

Figure 7: Samples $\tilde{x} = F^{-1}(z_s, \tilde{z}_n)$ with logit activations $z_s$ taken from original image and $\tilde{z}_n$ obtained by linearly interpolating from the original nuisance $z_n$ (first row) to the nuisance of a target example $z_n^*$ (last row upper block). The used target example is shown at the bottom. When training with cross-entropy, virtually any image can be turned into any class without changing the logits $z_s$, illustrating strong vulnerability to invariance-based adversaries. Yet, training with independence cross-entropy solves the problem and interpolations between nuisances $z_n$ and $z_n^*$ preserve the semantic content of the image.

A classifier trained on the nuisance variables of the cross-entropy trained model performs even better than the logit classifier. Yet, a classifier on the nuisances of the independence cross-entropy trained model is performing poorly (Table 2 in appendix D). This indicates little class-specific information in the nuisances $z_n$, as intended by our objective function. Note also that this inability of the nuisance classifier to decode class-specific information is not due to it being hard to read out from $z_n$, as this would be revealed by the metameric sampling attack (see Figure 7).

| % Error | $\mathbf{D_{Train}}$ | $\mathbf{D_{Adv}}$ |
|---|---|---|
| (a) CE   ResNet | 00.00 | 73.80 |
| (a) CE   fi-RevNet | 00.00 | 57.09 |
| (a) iCE   fi-RevNet | 00.02 | 34.73 |
| **(a) Difference** | **00.02** | **38.33** |
| (b) CE   ResNet | 00.00 | 87.83 |
| (b) CE   fi-RevNet | 00.18 | 73.71 |
| (b) iCE   fi-RevNet | 00.53 | 59.99 |
| **(b) Difference** | **00.53** | **27.84** |

Figure 8: shiftMNIST experiments. **(a):** Binary shiftMNIST, where the class is additionally encoded with a location-based binary code on the left border of the image (highlighted with red circles). The shifted adversarial test distribution does not have the binary class encoding. **(b):** Texture shiftM-NIST, where the class is additionally encoded in background texture type. The texture-class coupling is randomized in the shifted adversarial test distribution. **Right:** Results of CE-trained ResNet, fully invertible RevNet and iCE-trained fully invertible RevNet. The CE-based models build excessive invariance with respect to the digit identity on $\mathcal{D}_{train}$ and fail on $\mathcal{D}_{Adv}$. *Difference* denotes the largest improvement between CE-trained and iCE-trained model. The iCE model is more resilient to removing informative features, and reduces the error on $\mathcal{D}_{Adv}$ up to 38%.

**shiftMNIST: Benchmarking adversarial distribution shift.** To further test the efficacy of our proposed independence cross-entropy, we introduce a simple, but challenging new dataset termed *shiftMNIST* to test classifiers under adversarial distribution shifts $\mathcal{D}_{Adv}$. The dataset is based on vanilla MNIST, augmented by introducing additional, highly predictive features at train time that are randomized or removed at test time. Randomization or removal ensures that there are no synergy effects between digits and planted features under $\mathcal{D}_{Adv}$. This setup allows us to reduce mutual information between category and the newly introduced feature in a targeted manner. **(a)** Binary shiftMNIST is vanilla MNIST augmented by coding the category for each digit into a single binary pixel scheme. The location of the binary pixel reveals the category of each image unambigiously, while only minimally altering the image's appearance.

At test time, the binary code is not present and the network can not rely on it anymore. **(b)** Textured shiftMNIST introduces textured backgrounds for each digit category which are patches sampled from the describable texture dataset (Cimpoi et al., 2014).

At train time the same type of texture is underlayed each digit of the same category, while texture types across categories differ. At test time, the relationship is broken and texture backgrounds are paired with digits randomly, again minimizing the mutual information between background and label in a targeted manner. See Figure 8 for examples[1].

It turns out that this task is indeed very hard for standard classifiers and their tendency to become excessively invariant to semantically meaningful features, as predicted by our theoretical analysis. When trained with cross-entropy, ResNets and fi-RevNets make zero errors on the train set, while having error rates of up to 87% on the shifted test set. This is striking, given that e.g. in binary shiftMNIST, only one single pixel is removed under $\mathcal{D}_{Adv}$, leaving the whole image almost unchanged. When applying our independence cross-entropy, the picture changes again. The errors made by the network improve by up to almost 38% on binary shiftMNIST and around 28% on textured shiftMNIST. This highlights the effectiveness of our proposed loss function and its ability to minimize catastrophic failure under severe distribution shifts exploiting excessive invariance.

## 5 RELATED WORK

**Adversarial examples.** Adversarial examples often include $\epsilon$-norm restrictions (Szegedy et al., 2013), while (Gilmer et al., 2018a) argue for a broader definition to fully capture the implications for security. The $\epsilon$-adversarial examples have also been extended to $\epsilon$-feature adversaries (Sabour et al., 2016), which are equivalent to our approximate metameric sampling attack. Some works (Song et al., 2018; Fawzi et al., 2018) consider unrestricted adversarial examples, which are closely related to invariance-based adversarial vulnerability. The difference to human perception revealed by adversarial examples fundamentally questions which statistics deep networks use to base their decisions (Jo & Bengio, 2017; Tsipras et al., 2019).

**Relationship between standard and bijective networks.** We leverage recent advances in reversible (Gomez et al., 2017) and bijective networks (Jacobsen et al., 2018; Ardizzone et al., 2019; Kingma & Dhariwal, 2018) for our analysis. It has been shown that ResNets and iRevNets behave similarly on various levels of their representation on challenging tasks (Jacobsen et al., 2018) and that iRevNets as well as Glow-type networks are related to ResNets by the choice of dimension splitting applied in their residual blocks (Grathwohl et al., 2019). Perhaps unsurprisingly, given so many similarities, ResNets themselves have been shown to be provably bijective under mild conditions (Behrmann et al., 2018b). Further, excessive invariance of the type we discuss here has been shown to occur in non residual-type architectures as well (Gilmer et al., 2018b; Behrmann et al., 2018a). For instance, it has been observed that up to 60% of semantically meaningful input dimensions on the adversarial spheres problem are learned to be ignored, while retaining virtually perfect performance (Gilmer et al., 2018b). In summary, there is ample evidence that RevNet-type networks are closely related to ResNets, while providing a principled framework to study widely observed issues related to excessive invariance in deep learning in general and adversarial robustness in particular.

**Information theory.** The information-theoretic view has gained recent interest in machine learning due to the information bottleneck (Tishby & Zaslavsky, 2015; Shwartz-Ziv & Tishby, 2017; Alemi et al., 2017) and usage in generative modelling (Chen et al., 2016; Hjelm et al., 2019). As a consequence, the estimation of mutual information (Barber & Agakov, 2003; Alemi et al., 2018; Achille & Soatto, 2018; Belghazi et al., 2018) has attracted growing attention. The concept of group-wise independence between latent variables goes back to classical independent subspace analysis (Hyvärinen & Hoyer, 2000) and received attention in learning unbiased representations, e.g. see the Fair Variational Autoencoder (Louizos et al., 2015). Furthermore, extended cross-entropy losses via entropy terms (Pereyra et al., 2017) or minimizing predictability of variables (Schmidhuber, 1991) has been introduced for other applications. Our proposed loss also shows similarity to the GAN loss (Goodfellow et al., 2014). However, in our case there is no notion of real or fake samples, but exploring similarities in the optimization are a promising avenue for future work.

---

[1]Link to code and dataset: https://github.com/jhjacobsen/fully-invertible-revnet

## 6 CONCLUSION

Failures of deep networks under distribution shift and their difficulty in out-of-distribution generalization are prime examples of the limitations in current machine learning models. The field of adversarial example research aims to close this gap from a robustness point of view. While a lot of work has studied $\epsilon$-adversarial examples, recent trends extend the efforts towards the unrestricted case. However, adversarial examples with no restriction are hard to formalize beyond testing error. We introduce a reverse view on the problem to: (1) show that a major cause for adversarial vulnerability is excessive invariance to semantically meaningful variations, (2) demonstrate that this issue persists across tasks and architectures; and (3) make the control of invariance tractable via fully-invertible networks.

In summary, we demonstrated how a bijective network architecture enables us to identify large adversarial subspaces on multiple datasets like the adversarial spheres, MNIST and ImageNet. Afterwards, we formalized the distribution shifts causing such undesirable behavior via information theory. Using this framework, we find one of the major reasons is the insufficiency of the vanilla cross-entropy loss to learn semantic representations that capture all task-dependent variations in the input. We extend the loss function by components that explicitly encourage a split between semantically meaningful and nuisance features. Finally, we empirically show that this split can remove unwanted invariances by performing a set of targeted invariance-based distribution shift experiments.

## 7 ACKNOWLEDGEMENTS

We thank Ryota Tomioka for spotting a mistake in the proof for Theorem 6. We thank thank the anonymous reviewers, Ricky Chen, Will Grathwohl and Jesse Bettencourt for helpful comments on the manuscript. We gratefully acknowledge the financial support from the German Science Foundation for the CRC 1233 on "Robust Vision" and RTG 2224 "$\pi^3$: Parameter Identification - Analysis, Algorithms, Applications"

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

## A    Semantic and Nuisance Variation on Adversarial Spheres

**Example 7** (Semantic and nuisance on Adversarial Spheres (Gilmer et al., 2018b))**.** *Consider classifying inputs $x$ from two classes given by radii $R_1$ or $R_2$. Further, let $(r, \phi)$ denote the spherical coordinates of $x$. Then, any perturbation $\Delta x$, $x^* = x + \Delta x$ with $r^* \neq r$ is semantic. On the other hand, if $r^* = r$ the perturbation is a nuisance with respect to the task of discriminating two spheres.*

In this example, the max-margin classifier $D(x) = sign\left(\|x\| - \frac{R_1+R_2}{2}\right)$ is invariant to any nuisance perturbation, while being only sensitive to semantic perturbations. In summary, the transform to spherical coordinates allows to linearize semantic and nuisance perturbations. Using this notion, invariance-based adversarial examples can be attributed to perturbations of $x^* = x + \Delta x$ with following two properties

1. Perturbed sample $x^*$ stays in the pre-image $\{x^* \in \mathbb{R}^d \mid D(x^*) = D(x)\}$ of the classifier

2. Perturbation $\Delta x$ is semantic, as $o(x) \neq o(x + \Delta x)$.

Thus, the failure of the classifier $D$ can be thought of a mis-alignment between its invariance (expressed through the pre-image) and the semantics of the data and task (expressed by the oracle).

**Example 8** (Mis-aligned classifier on Adversarial Spheres)**.** *Consider the classifier*

$$D(x) = sign\left(\|x_{1,\ldots,d-1}\| - \frac{R_1 + R_2}{2}\right), \tag{7}$$

*which computes the norm of $x$ from its first $d-1$ cartesian-coordinates. Then, $D$ is invariant to a semantic perturbation with $\Delta r = R_2 - R_1$ if only changes in the last coordinate $x_d$ are made.*

We empirically evaluate the classifier in equation 7 on the spheres problem (10M/2M samples setting (Gilmer et al., 2018b)) and validate that it can reach perfect classification accuracy. However, by construction, perturbing the invariant dimension $x_d^* = x_d + \Delta x_d$ allows us to move all samples from the inner sphere to the outer sphere. Thus, the accuracy of the classifier drops to chance level when evaluating its performance under such a distributional shift.
To conclude, this underlines how classifiers with optimal performance on finite samples can exhibit non-intuitive failure modes due to excessive invariance with respect to semantic variations.

## B    Approximate Gradient-based Metameric Samples

We use a standard Imagenet pre-trained Resnet-154 as provided by the torchvision package (Paszke et al., 2017) and choose a logit percept $\mathbf{y} = G(\mathbf{x})$ that can be based on any seed image. Then we optimize various images $\tilde{x}$ to be metameric to $\mathbf{x}$ by simply minimizing a mean squared error loss of the form:

$$\mathcal{L}_{\text{MSE}}(G(x), G(\tilde{x})) = \frac{1}{K}\sum_{k=1}^{K}(G(x)_k - G(\tilde{x})_k)^2 \tag{8}$$

in the 1000-dimensional semantic logit space via stochastic gradient descent. We optimize with Adam in Pytorch default settings and a learning rate of 0.01 for 3000 iterations. The optimization thus takes the form of an adversarial attack targeting all logit entries and with no norm restriction on the input distance. Note that our metameric sampling attack in bijective networks is the analytic reverse equivalent of this attack. It leads to the exact solution at the cost of one inverse pass instead of an approximate solution here at the cost of thousands of gradient steps.

### B.1 Additional Batch of Metameric Samples

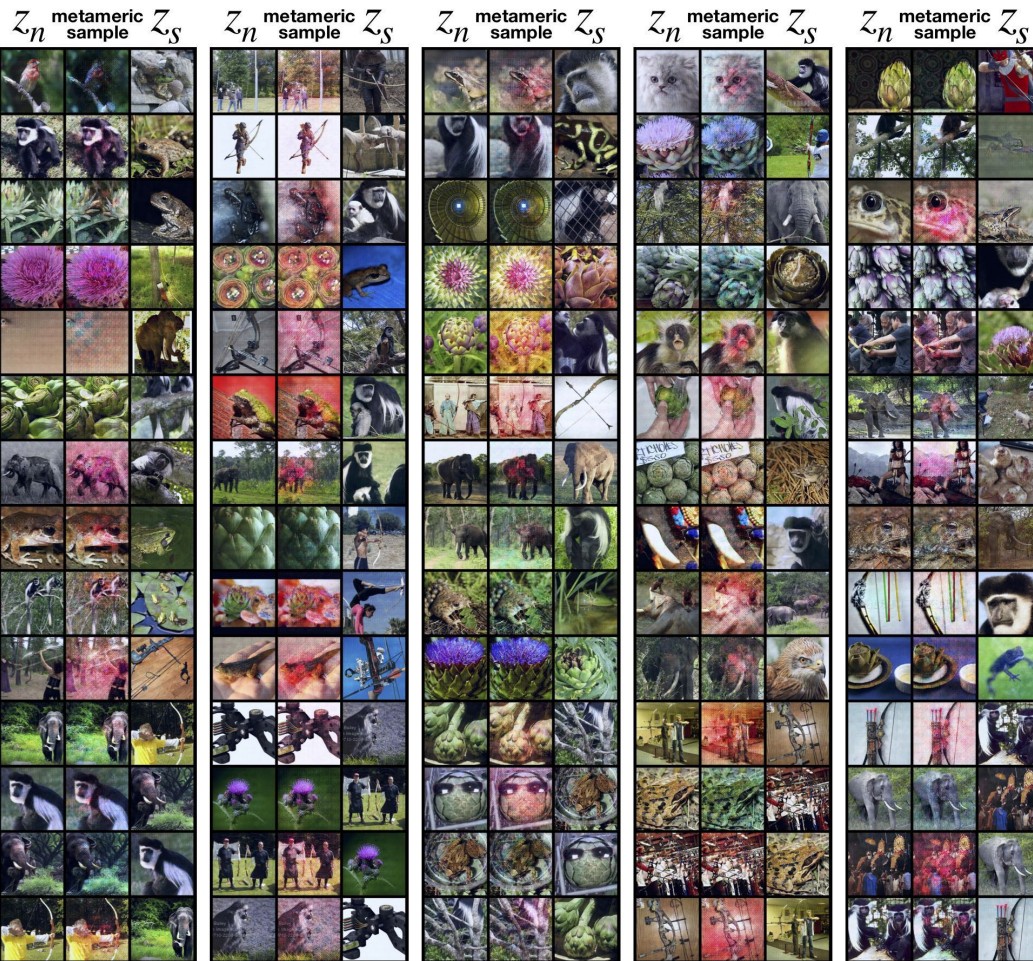

Figure 9: Here we show a batch of randomly sampled metamers from our ImageNet-trained fully invertible RevNet-48. The quality is generally similar, sometimes colored artifacts appear.

## C Information Theory

Computing mutual information is often intractable as it requires the joint probability $p(x, y)$, see (Cover & Thomas, 2006) for an extensive treatment of information theory. However, following variational lower bound can be used for approximation, see (Barber & Agakov, 2003).

**Lemma 9** (Variational lower bound on mutual information). *Let $X, Y$ be random variables with conditional density $p(y|x)$. Further, let $q_\theta(y|x)$ be a variational density depending on parameter $\theta$. Then, the lower bound*

$$I(Y; X) = h(Y) - h(Y|X) = h(Y) + \mathbb{E}_X \mathbb{E}_{Y|X} \log q_\theta(y|x) + \mathbb{E}_X(p(y|x) \| q_\theta(y|x))$$
$$\geq h(Y) + \mathbb{E}_X \mathbb{E}_{Y|X} \log q_\theta(y|x)$$

*holds with equality if $p(y|x) = q_\theta(y|x)$.*

While above lower bound removes the need for the computation of $p(y|x)$, estimating the expectation $\mathbb{E}_{Y|X}$ still requires sampling from it. Using this bound, we can now state the effect of the nuisance classification loss.

**Lemma 10** (Effect of nuisance classifier). *Define semantics as $z_s = F_\theta(x)_{1,...,C}$ and nuisances as $z_n = F_\theta(x)_{C+1,...,d}$, where $(x, y) \sim \mathcal{D}$. Then, the nuisance classification loss yields*

(i) **Minimization of lower bound on $I_\mathcal{D}(y; z_n)$:** $\theta^* = \arg\min_\theta \mathcal{L}_{nCE}(\theta, \theta_{nc}^*)$ minimizes $I_{\theta_{nc}^*}(y; z_n)$, where $I_{\theta_{nc}^*}(y; z_n) \leq I_\mathcal{D}(y; z_n)$ and $\theta_{nc}^* = \arg\max_{\theta_2} \mathcal{L}_{nCE}(\theta, \theta_{nc})$.

(ii) **Maximization to tighten bound on $I_\mathcal{D}(y; z_n)$:** *Under a perfect model of the conditional density, $D_{\theta_{nc}^*}(z_n) = p(y|z_n)$, it holds $I_{\theta_{nc}^*}(y; z_n) = I_\mathcal{D}(y; z_n)$.*

*Proof.* To proof above result, we need to draw the connection to the variational lower bound on mutual information from Lemma 9. Let the nuisance classifier $D_{\theta_{nc}}(z_n)$ model the variational posterior $q_{\theta_{nc}}(y|z_n)$. Then we have the lower bound

$$I(y; z_n) \geq h(y) + \mathbb{E}_{z_n}\mathbb{E}_{y|z_n} \log D_{\theta_{nc}}(z_n) =: I_{\theta_{nc}}(y; z_n). \tag{9}$$

From Lemma 9 follows, that if $D_{\theta_{nc}}(z_n) = p(y|z_n)$, it holds $I(y; z_n) = I_{\theta_{nc}}(y; z_n)$. Hence, the nuisance classifier needs to model the conditional density perfectly.

Estimating this bound via Monte Carlo simulation requires sampling from the conditional density $p(y|z_n)$. Following (Alemi et al., 2017), we have the Markov property $y \leftrightarrow x \leftrightarrow z_n$ as labels $y$ interact with inputs $x$ and representation $z_n$ interacts with inputs $x$. Hence,

$$p(y|z_n)p(z_n) = p(y, z_n)$$
$$= \int_\mathcal{X} p(x, y, z_n) dx$$
$$= \int_\mathcal{X} p(z_n|x, y)p(y|x)p(x) dx$$
$$= \int_\mathcal{X} p(z_n|x)p(y|x)p(x) dx$$
$$= \mathbb{E}_x[p(z_n|x)p(y|x)].$$

Including above and assuming $F_\theta(x) = z_n$ to be a deterministic function, we have

$$\mathbb{E}_{z_n}\mathbb{E}_{y|z_n} \log D_{\theta_{nc}}(z_n) = \mathbb{E}_x\mathbb{E}_{y|x}\mathbb{E}_{z_n|x} \log D_{\theta_{nc}}(z_n) = \mathbb{E}_x\mathbb{E}_{y|x} \log D_{\theta_{nc}}(z_n).$$

$\square$

**Lemma 11** (Effect of MLE-term). *Define semantics as $z_s = F_\theta(x)_{1,...,C}$ and nuisances as $z_n = F_\theta(x)_{C+1,...,d}$, where $(x, y) \sim \mathcal{D}$. Then, the MLE-term in equation 6 together with cross-entropy on the semantics*

$$\theta^* = \arg\min_\theta \mathcal{L}_{sCE}(\theta) + \mathcal{L}_{MLE_n}(\theta)$$

*minimizes the mutual information $I(z_s; z_n)$.*

*Proof.* Let $\tilde{z}_s = softmax(z_s)$. Then minimizing the loss terms $\mathcal{L}_{sCE}$ and $\mathcal{L}_{MLE_n}$ is a maximum likelihood estimation under the factorial prior

$$p(\tilde{z}_s, z_n) = p(\tilde{z}_s)p(z_n) \tag{10}$$
$$= Cat((\tilde{z}_s)_1, \ldots, (\tilde{z}_s)_C) \prod_{k=1}^{d-C} p_k(z_n)_k, \tag{11}$$

where $Cat$ is a categorical distribution. As $softmax$ is shift-invariant, $softmax(x + c) = softmax(x)$, above factorial prior for $\tilde{z}_s$ and $z_n$ yields independence between logits $z_s$ and $z_n$ up to a constant $c$. Finally note, the $log$ term and summation in $\mathcal{L}_{MLE_n}$ and $\mathcal{L}_{CE}$ is re-formulation for computational ease but does not change its minimizer as the logarithm is monotone. $\square$

### C.1 PROOF OF THEOREM 6

From the assumptions follows $I_{\mathcal{D}_{Adv}}(y; z_n) = 0$. Furthermore, we have the assumption

$$I_{\mathcal{D}_{Adv}}(y; z_s|z_n) \leq I_{\mathcal{D}_{Adv}}(z_s; y),$$

excluding synergetic effects in the interaction information (Ghassami & Kiyavash, 2017). By information preservation under homeomorphisms (Kraskov et al., 2004) and the chain rule of mutual information (Cover & Thomas, 2006), we have

$$
\begin{aligned}
I_{\mathcal{D}_{Adv}}(y; x) &= I_{\mathcal{D}_{Adv}}(y; z_s, z_n) \\
&= I_{\mathcal{D}_{Adv}}(y; z_n) + I_{\mathcal{D}_{Adv}}(y; z_s | z_n) \\
&\leq I_{\mathcal{D}_{Adv}}(y; z_s).
\end{aligned}
$$

As $z_s = F(x)_{1,\dots,C}$ is obtained by the deterministic transform $F$, by the data processing inequality (Cover & Thomas, 2006) we have the inequality $I_{\mathcal{D}_{Adv}}(y; x) \geq I_{\mathcal{D}_{Adv}}(y; z_s)$. Thus, the claimed equality must hold.

## C.2 MUTUAL INFORMATION BOUNDED

**Remark 12.** *Since our goal is to maximize the mutual information $I(y; z_s)$ while minimizing $I(y; z_n)$, we need to ensure that this objective is well defined as mutual information can be unbounded from above for continuous random variables. However, due to the data processing inequality (Cover & Thomas, 2006) we have $I(y; z_n) = I(y; F_\theta(x)) \leq I(y; x)$. Hence, we have a fixed upper bound given by our data $(x, y)$. Compared to (Belghazi et al., 2018) there is thus no need for gradient clipping or a switch to the bounded Jensen-Shannon divergence as in (Hjelm et al., 2019) is not necessary.*

## D  TRAINING AND ARCHITECTURAL DETAILS

All experiments were based on a fully invertible RevNet model with different hyperparameters for each dataset. For the spheres experiment we used Pytorch (Paszke et al., 2017) and for MNIST, as well as Imagenet Tensorflow (Abadi et al., 2016).

### D.1  SPHERES EXPERIMENTS

The network is a fully connected fully invertible RevNet. It has 4 RevNet-type ReLU bottleneck blocks with additive couplings and uses no batchnorm. We train it via cross-entropy and use the Adam optimizer (Kingma & Ba, 2014) with a learning rate of 0.0001 and otherwise default Pytorch settings. The nuisance classifier is a 3 layer ReLU network with 1000 hidden units per layer.

We choose the spheres to be 100-dimensional, with $R_1 = 1$ and $R_2 = 10$, train on 500k samples for 10 epochs and then validate on another 100k holdout set. We achieve 100% train and validation accuracy for logit and nuisance classifier.

### D.2  MNIST EXPERIMENTS

We use a convolutional fully invertible RevNet with additional actnorm and invertible 1x1 convolutions between each layer as introduced in Kingma & Dhariwal (2018). The network has 3 stages, after which half of the variables are factored out and an invertible downsampling, or squeezing (Dinh et al., 2017; Jacobsen et al., 2018) is applied. The network has 16 RevNet blocks with batch norm per stage and 128 filters per layer. We also dequantize the inputs as is typically done in flow-based generative models.

The network is trained via Adamax (Kingma & Ba, 2014) with a base learning rate of 0.001 for 100 epochs and we multiply the it with a factor of 0.2 every 30 epochs and use a batch size of 64 and l2 weight decay of 1e-4. For training we compare vanilla cross-entropy training with our proposed independence cross-entropy loss. To have a more balanced loss signal, we normalize $\mathcal{L}_{nCE}$ by the number of input dimensions it receives for the maximization step. The nuisance classifier is a fully-connected 3 layer ReLU network with 512 units. As data-augmentation we use random shifts of 3 pixels. For classification errors of the different architectures we compare, see Table 2.

### D.3  IMAGENET EXPERIMENTS

We use a convolutional fully invertible RevNet with 4 stages, 4 RevNet blocks per stage and invertible downsampling after each stage, as well as two invertible downsamplings on the input of the

| MNIST | **SOTA** | **LeNet** | **CE** | **iCE** (ours) | **CE** | **iCE** (ours) |
|---|---|---|---|---|---|---|
| Readout | Logit | Logit | Logit | Logit | Nuisance | Nuisance |
| % Test Error | 0.21 | 1.70 | 0.39 | 0.38 | 0.34 | 27.70 |
| % Train Error | - | - | 0.00 | 0.37 | 0.00 | 40.21 |

Table 2: Results comparing cross-entropy training (CE) with independence cross-entropy training (iCE) from Definition 5 and two architectures from the literature. The accuracy of the logit classifiers is on par for the CE and iCE networks, but the train error is higher for CE compared to test error, indicating less overfitting for iCE. Further, a classifier independently trained on the nuisance variables is able to reach even smaller error than on the logits for CE, but just 27.70% error for iCE, indicating that we have successfully removed most of the information of the label from the nuisance variables and fixed the problem of excessive invariance to semantically meaningful variability with no cost in test error.

network. The first three stages consist of additive and the last of affine coupling layers. After the final layer we apply an orthogonal 2D DCT type-II to all feature maps and read out the classes in the low-pass components of the transformation. This effectively gives us an invertible global average pooling and makes our network even more similar to ResNets, that always apply global average pooling on their final feature maps. We train the network with momentum SGD for 128 epochs, a batch size of 480 (distributed to 6 GPUs), a base learning rate of 0.1, which is reduced by a factor of 0.1 every 32 epochs. We apply momentum of 0.9 and l2 weight decay of 1e-4.

