# OpenReview forum: "Excessive Invariance Causes Adversarial Vulnerability"
_ICLR.cc/2019/Conference_

### Official Review · AnonReviewer1 · 2018-11-01
**The ideas are appealing and should enventually lead to fine contributions but the paper is disbalanced with wrong detail distribution**

**Rating:** 6
**Confidence:** 2

**Review:**

The paper focuses on adversarial vulnerability of neural networks, and more specifically on perturbation-based versus invariance-based adversarial examples and how using bijective networks (with so-called metameric sampling) may help overcoming issues related to invariance. The approach is used to get around insufficiencies of cross-entropy-based information-maximization, as illustrated on experiments where the proposed variation on CE outperforms CE.

While I am not a neural network expert, I felt that the ideas developed in the paper are worthwhile and should eventally lead to useful contributions and be published. This being said, I did not find the paper in its present form to be fit for publication in a high-tier conference or journal. The main reason for this is the disbalance between the somehow heavy and overly commented first four pages (especially in Section 2) contrasting with the surprisingly moderate level of detail when it comes to bijective networks, supposedly the heart of the actual original contribution. To me this is severely affecting the overall quality of the paper. The contents of sections 3 and 4 seem relevant, but I struggled find out what precisely is the main contribution in the end, probably because of the lack of detail on bijective networks mentioned before. Again, I am not an expert, and I will indicate that in the system of course, but while I cannot completely judge all aspects of the technical relevance and the originality of the approach, I am fairly convinced that the paper deserves to be substantially revised before it can be accepted for publication.

Edit: After paper additions I am changing my score to a 6.

---

> ### Author Response · Authors · 2018-11-10
> **Thoroughly revised manuscript uploaded**
>
>
> --------------------------------------------------------
>
> We thank you very much for acknowledging our work being appealing and our contributions being publication-worthy.
> We also thank you for your thoughts and comments on the structure of the manuscript.
>
> --------------------------------------------------------
>
> Q: Overly commented first pages, imbalanced with section 3 and 4.
>
> We have done our best to fix this and have substantially revised the paper. We removed large portions of section 2 and added it to the appendix, we added additional details about bijective networks, re-structured section 3 and 4 and added another experiment to emphasise our main contributions more. Finally, we have adjusted the abstract and contributions in the introduction accordingly.
>
> --------------------------------------------------------
>
> Q: Lacking detail on bijective network.
>
> The main components we are using are based on Real-NVP[1]/Glow[2] and iRevNet[3] networks, which are widely known and cited in the paper, so we decided not to put too much focus on their details.
> However, in the revision we have added some additional details, for instance, we have added figure 3 that explains the architecture we are using.
>
> [1] Dinh, Laurent, Jascha Sohl-Dickstein, and Samy Bengio. "Density estimation using Real NVP."
> [2] Kingma, Diederik P., and Prafulla Dhariwal. "Glow: Generative flow with invertible 1x1 convolutions."
> [3] Jacobsen, Jörn-Henrik, Arnold Smeulders, and Edouard Oyallon. "i-RevNet: Deep Invertible Networks."
> --------------------------------------------------------
>
> Please let us know if you have any more comments or concerns!
>
> Thank you once again.

---

### Official Review · AnonReviewer2 · 2018-11-05
**An interesting problem with an unconvincing solution**

**Rating:** 6
**Confidence:** 4

**Review:**

This paper studies a new perspective on why adversarial examples exist in machine learning -- instead of seeing adversarial examples as the result of a classifier being sensitive to changes in irrelevant information (aka nuisance), the authors see them as the result of a classifier being invariant to changes in relevant (aka semantic) information. They show how to efficiently find such adversarial examples in bijective networks. Moreover, they propose to modify the training objective so that the bijective networks could be more robust to such attacks.

Pros:
 -- clarity is good (except for a few places, e.g. no definition of F(x)_i in Definition 1; Page 6 "three ways forward" item 3: I(y;z_n|z_s) = I(y;z_s) should be I(y;z_n|z_s) = I(y;z_n).)
 -- the idea is original to the best of my knowledge
 -- the mathematical motivation is sound
 -- Figure 6 seems to show that the proposed defense works on MNIST (However, would you provide more details on how you interpolated z_n? Moreover, what do the images generated with z_s from one input and z_n from another input look like (in your method)?)

Cons:
 -- scope: as all the presented problems and solutions assume bijective mapping, I wonder how is it relevant to the traditional perspective of adversarial attack and defense? It seems to me that the contribution of this paper is identifying a problem of bijective networks and then proposing a solution, thus its significance is restricted.
 -- method: while the mathematical motivation is sound, I'm not sure if the proposed training objective can achieve that goal. To elaborate, I see problems with both terms added in the proposed loss function:
 (a.) for the objective of maximizing the cross entropy of the nuisance classifier, it is possible that I(y;z_n) is not reduced, but rather the information about y is encoded in a way that the nuisance classifier is not able to decode, similar to what happens in a one-way function (for example, see https://en.wikipedia.org/wiki/Cryptographic_hash_function ). In the MNIST experiments, the nuisance classifier is a three-layer MLP, which may be too weak and susceptible to information concealing.
 (b.) for the objective of maximizing the likelihood of a factorized model of p(z_s, z_n), I don't see how optimizing it would reduce I(z_s; z_n). In general, even if z_s and z_n are strongly correlated, one can still fit such a factorized model. This only ensures that I(Z_s; Z_n) = 0 for Z_s, Z_n *sampled from the model*, but does not necessarily reduce I(z_s; z_n) for z_s, z_n *used to train the model*. The discrepancy between p(Z_s, Z_n) and p(z_s, z_n) could be huge, in which case one has the model misspecification problem which is another topic.
 (c.) a side question: why is the MLE objective using likelihood rather than log likelihood? Since the two cross entropy losses are similar to log likelihood, I feel there is a mismatch here.

----------------------------------------
AFTER REBUTTAL:

Thanks for your reply to my comments. The new revision has improved clarity and provided new supporting evidences. I would like to raise my rating to 6.

That being said, (as you agreed) the link from the conceptual goal to the proposed objective has mostly empirical support. Therefore I hope it may encourage future investigation on when and why the proposed objective is successful in achieving the conceptual goal.

---

> ### Author Response · Authors · 2018-11-10
> **Added new experiments on non-bijective networks and proposed objective alongside thorough revision of manuscript**
>
>
> --------------------------------------------------------
>
> We are glad that you find most of our major contributions original, interesting, clear and mathematically sound.
> We also thank you for your thoughtful questions and comments, we address them below.
>
> --------------------------------------------------------
>
> Q: How are findings related to non-bijective networks?
> --
> Thank you for bringing this up, we have revised the manuscript to answer this important question very clearly to show our identified problems, analysis and conclusion are not limited to bijective networks.
> We summarize below.
>
> ----------
> -- Our identified problem of excessive invariance occurs in many other networks as well.
> ----------
>
> We have added results on the gradient-based equivalent of our analytic metameric sampling attack to the paper. We match the logit vector of one image with the logits of another image via gradient-based optimization and no norm-based restriction on the input. We do so on an ImageNet-trained state of the art ResNet-154 and see that the problem we have identified in bijective nets is the same here, if not worse as the metameric samples look even cleaner. Qualitative results are added to figure 5.
>
> Besides that, multiple papers have observed excessive invariance. On the adversarial spheres problem [1], for instance, the authors show their quadratic network does almost perfectly well while ignoring up to 60% of *semantically meaningful* input dimensions. Another line of work has also shown that similar behavior can appear in ReLU networks as well [2].
>
> We have also added an additional set of experiments to the revised manuscript that shows how cross-entropy trained ResNets fail badly under distribution shifts that exploit their excessive invariance, giving another piece of evidence that our findings are not limited to bijective networks, but applicable to the most successful deep network architecture around as well.
>
> ----------
> -- There is a close relationship between bijective nets and SOTA architectures.
> ----------
>
> Bijective networks are closely related to ResNets, they are in fact provably bijective under mild assumptions, as shown by a recent publication [3]. Further, it has been shown that ResNets and RevNet-type networks differ only in their dimension splitting scheme from one another [4]. And finally, bijective iRevNets have been shown to have many equivalent progressive properties to ResNets throughout the layers of their learned representation [5].
>
> In summary, there is ample evidence, that bijective RevNet-type networks are not the reason for the problems we observe, but rather extremely similar to ResNets, the de-facto state-of-the-art architecture, while providing a powerful framework to study and combat problems like excessive invariance.
>
> [1] Gilmer, Justin, et al. "Adversarial spheres."
> [2] Behrmann, Jens, et al. "Analysis of Invariance and Robustness via Invertibility of ReLU-Networks."
> [3] Behrmann, Jens, David Duvenaud, and Jörn-Henrik Jacobsen. "Invertible Residual Networks."
> [4] Grathwohl, Will, et al. "FFJORD: Free-form Continuous Dynamics for Scalable Reversible Generative Models."
> [5] Jacobsen, Jörn-Henrik, Arnold Smeulders, and Edouard Oyallon. "i-RevNet: Deep Invertible Networks."

---

> > ### Author Response · Authors · 2018-11-10
> > **Part II**
> >
> >
> > ---------------------------------------------------------
> >
> > Q: Can the training objective achieve its goal?
> >
> > ----------
> > -- (a) The nuisance classifier is not powerful enough to decode y from z_n.
> > ----------
> >
> > This is indeed a common problem when formulating a bound this way and it is the same problem GANs face. However, in practice, GANs often work and we also find that the nuisance classifier does indeed do its job, one could even validate this post-hoc by training a more powerful nuisance classifier to confirm it.
> >
> > Additionally, we also have metameric sampling as a validation method. If the information about the class is only hidden in z_n, but not removed, then metameric sampling would reveal this. Replacing z_n of one category with a z_n from another category would then change the category of the reconstruction, but we see that this is not happening when applying our loss. Thus, we conclude that the objective is successful, albeit it having its challenges.
> >
> > ----------
> > -- (b) The factorial maximum likelihood objective does not lead to independence.
> > ----------
> >
> > We agree, that there is no guarantee that the loss will lead to full independence, but it does encourage it at least.
> > On the other hand, our evaluation method (metameric sampling) is not based on samples from the model but is based on the activations of real data points. Thus, according to your argumentation, this sampling method would reveal strong dependencies between the subspaces. In practice, we see this is not the case, as shown in figure 7 on the right, where combinations of z_n from one class and z_s from another do indeed lead to a change of nuisance/style in the original image, but not to a change of category. Empirically this means our objective was successful and most of the label information has been removed from z_n.
> >
> > To further analyze the objective, we have added another experiment to assess if it can successfully defend against targeted distribution shifts as considered in Theorem 6.
> > We introduce a new dataset termed shiftMNIST, it augments MNIST with additional highly predictive features at train time and removes or randomizes those features at test time, while leaving the digits themselves as the stable predictive variable.
> >
> > Our experiments reveal, that the baseline cross-entropy trained ResNet and fiRevNet fail badly on these problems, while our proposed loss reduces the error under such distribution shift up to 38%. This provides more evidence that our proposed objective does achieve its goal in practice.
> >
> > ----------
> >
> > In summary, we do agree that the lower bound and the maximum likelihood objectives have their respective issues and we added some discussion on this to the manuscript. However, in practice, the metameric samples and our additional distribution shift experiments show that the loss does, in fact, work as intended, making it a promising way forward.
> >
> > ---------------------------------------------------------
> >
> > Q: What do the images generated with z_s from one input and z_n from another input look like (in your method)?
> > --
> > Those images (the metameric samples) are already shown in the last row in the top block of figure 7, we have adapted the figure and added some more description to it, to make everything more clear.
> > In the baseline the metameric samples are adversarial examples, meaning one can turn any image into any class without changing the logits at all. With our objective (shown on the right side), this is not possible anymore as keeping z_s fixed and exchanging z_n only affects the style of the image, not its class-specific content. The objective has achieved its goal and successfully defended against the metameric sampling attack.
> >
> > ---------------------------------------------------------
> >
> > Minor:
> > We have fixed the typos and added the log to the MLE objective, thank you.
> >
> > ---------------------------------------------------------
> >
> > Thank you once again for the detailed review, we were able to significantly improve the manuscript based on it.
> > We have revised multiple parts, added new experiments and added discussions to answer your concerns.
> >
> > We hope we were able to answer everything to your satisfaction, please let us know if there are any more open points.
> >
> > Thank you once again!

---

> > > ### Author Response · Authors · 2018-11-29
> > > **Further concerns after revision?**
> > >
> > > Dear Reviewer2,
> > >
> > > we would be most grateful if you can let us know if there are any further concerns you have after considering the thoroughly revised manuscript, added experiments and answers above.

---

> > > > ### Comment · AnonReviewer2 · 2018-11-29
> > > > **Thanks for the rebuttal and revision. I would like to raise my score.**
> > > >
> > > > Dear Authors,
> > > >
> > > > Thanks for your reply to my comments. The new revision has improved clarity and provided new supporting evidences.
> > > >
> > > > That being said, (as you agreed) the link from the conceptual goal to the proposed objective has mostly empirical support. Therefore I hope it may encourage future investigation on when and why the proposed objective is successful in achieving the conceptual goal.
> > > >
> > > > Best,

---

> > > > > ### Author Response · Authors · 2018-12-04
> > > > > **Thank you for the discussion!**
> > > > >
> > > > > We were glad to see your positive feedback.
> > > > >
> > > > > Indeed we agree some open questions (summarized below in point (II)) remain. Yet, we hope that our efforts to prove the underlying principles of our objective sparks future analysis how/when our optimality assumptions (discussed below in point (I)) can be achieved and why the objective succeeds in our current setting.
> > > > > That being said, as pointed out above, the objective function itself is one out of 4 major contributions and therefore this analysis would be out of scope for the presented work.
> > > > >
> > > > > Thank you once again for the constructive discussion!
> > > > >
> > > > > ------------------------------
> > > > > (I) Optimality assumptions:
> > > > > - Lemma 8 (i) (Appendix A): CE- and MLE-term is Maximum Likelihood under a factorized prior p(z_s, z_n) = p(z_s) p(z_n). In the optimum, it thus holds I(z_s; z_n) = 0 as I(z_s; z_n) = KL(p(z_s, z_n) || p(z_s) p(z_n)).
> > > > > Furthermore, in the optimum I(y; z_s) = H(y) = const.
> > > > > - Lemma 8 (ii) and (iii) (Appendix A): If the lower bound is tight (nuisance classifier can decode all information about y in z_n), we minimize I(y; z_n) provably.
> > > > > ------------------------------
> > > > > (II) Achieving optimality / Possible alternatives:
> > > > > - Connecting independence of z_n and z_s with the model architecture: Due to information preservation, bijective networks are particularly suitable for our task, but other architectures could be considered.
> > > > > - Tightness of lower bounds: How tight are lower bounds given by a nuisance classifier or alternative lower bounds by the MINE estimator (Belghazi et al. 2018)?
> > > > > - Lack of alternatives: As I(y; z_n) is bounded (Remark 9, Appendix A.2), non-trivial (smaller than H(y)) upper bounds on I(y; z_n) are difficult and to the best of our knowledge, we are not aware of any.

---

### Official Review · AnonReviewer3 · 2018-11-06
**Very interesting ideas, could use a few additional experiments to be more convincing**

**Rating:** 7
**Confidence:** 4

**Review:**

This paper explores adversarial examples by investigating an invertible neural network. They begin by first correctly pointing out limitations with the commonly adopted "l_p adversarial example" definition in literature. The main idea involves looking at the preimage of different embeddings in the final layer of an invertible neural network. By training a classifier on top of the final embedding of the invertible network the authors are able to partition the final embedding into a set of "semantic variables", which are the components used for classification of the classifier, and a set of "nuisance variables" which are the complement of the logit variables. This partition allows the authors to define entire subspaces of adversarial images by holding the logit variables fixed and varying the nuisance variables, and applying the inverse to these modified embeddings. The authors are able to find many incorrectly classified images with this inversion technique. The authors then define a new loss which minimizes the mutual information between the nuisance variables and the predicted label.

I found the ideas in this paper quite interesting and novel. Starting with the toy problem of adversarial spheres is great, and it's convincing that the inversion technique can be used to find errors on this dataset even when the classification accuracy is (empirically) 100%. The resulting adversarial images generated by applying their technique are also quite interesting, and this is a cool interesting way to study the robustness of networks in non-iid settings.

The main weakness is on the evaluation of their proposed new training objective, and I have a few suggestions as to how to strengthen this evaluation. It would be very convincing to me if the authors could show that their new training objective increases robustness to distributional shift. A potential benchmark for distributional shift could be https://arxiv.org/abs/1807.01697 (or just picking a subset of these image corruptions). If the proposed objective shows improvement on this benchmark (or a related one) then this would be a solid contribution.

One question I have for the authors is how typical the behavior in Figure 4 is? For any fixing of the logits, are all/most metameric samples classifiable by a human oracle? That is do you ever get garbage images from this sampling process. Adding a collection of random samples to the Appendix to demonstrate typical behavior could help demonstrate this.

Edit: After paper additions I am changing my score to a 7.

---

> ### Author Response · Authors · 2018-11-10
> **Added new experiments on distribution shift and large batch of metameric samples**
>
>
> --------------------------------------------------------
>
> We thank you very much for acknowledging our work as interesting and novel, as well as for the appreciation of our developed methodologies.
>
> We answer your questions below.
>
> --------------------------------------------------------
>
> Q: Does the new training objective increase robustness to distributional shift?
> --
> Thank you for raising this point. To shed light on the effect of our loss under adversarial distribution shifts we have added new experiments on a dataset we introduce to precisely test our claims. We term the dataset shiftMNIST and designed it such that it follows distribution shifts D_Adv of the form we assumed for Theorem 6.
>
> Our results reveal, that our proposed loss does indeed reduce the errors under challenging distribution shifts up to 38% as compared to cross-entropy trained ResNets and RevNets, highlighting the efficacy of our proposed objective.
>
> Further, the results also show once again how badly standard networks can fail, even though in one task only one single pixel is removed, leaving the image semantics almost entirely unchanged. The results are one more piece of evidence for the insufficiency of cross-entropy based information maximization and the excessive invariance it may lead to in practice.
>
> We sincerely thank you for bringing this up.
>
> --------------------------------------------------------
>
> Q: What is the typical behavior of samples shown in Figure 4?
> --
> The metameric samples shown are representative and we have observed similar quality throughout the whole validation set, sometimes with slight colored artifacts though. We have added a large batch of metameric samples to the appendix to give the reader a better idea about their typical behavior.
>
> --------------------------------------------------------
>
> We believe your review have substantially improved the manuscript, thank you.

---

### Author Response · Authors · 2018-11-10
**Revision uploaded**

Dear Reviewers, we thank you very much for helping us to substantially improve the manuscript.

We have addressed all raised concerns either with additional experiments and results, with additional discussions in the manuscript or through other aspects of our revision.

We were delighted to see the positive reaction by all reviewers to our developed ideas and your suggestions and concerns greatly improved the paper. The new distribution shift experiments, as well as new results and discussion of non-bijective networks and their relationship to bijective ones, significantly increase the practical relevance of the work.

Given the tension between the positive comments to most of our contributions, the ratings and the fact that the main concerns are related to our proposed solution, we would like to point out that the developed training objective is only one out of four major contributions of the paper.

We list our updated contributions here again for clarity:

1 - We introduce an alternative viewpoint on adversarial examples, one of the major failures in modern machine learning algorithms, give a formal definition of it and show its practical relevance for commonly used architectures in the updated experiments and discussion.

2 - We build a competitive bijective ImageNet/MNIST classifier to tractably compute such adversarial examples exactly. Based on this, we provide what may be the first analytic adversarial attack method in the literature.

3 - We prove that a major reason for invariance-based vulnerability is the commonly used cross-entropy objective and show from an information-theoretic viewpoint what may be done to overcome this.

4 - We put our theoretical results into practice: based on bijective networks we introduce a practically useful loss and illustrate as a proof-of-concept that it largely overcomes the problem of excessive invariance, making it a promising way forward. Additionally, we have now included more quantitative experiments showing robustness to adversarial distribution shifts on a newly introduced benchmark.

In the revision we have:

-- Thoroughly revised and updated the whole manuscript to make all of our contributions more clear and incorporate all raised concerns.
-- Updated figures and descriptions and moved large parts of section 2 to the appendix to improve clarity.
-- Added an adversarial distribution shift benchmark to stress test our proposed objective and show its effectiveness in challenging settings.
-- Added new results on non-bijective networks for the metameric samples and the distribution shift experiments to show non-bijective networks have the same issues as the bijective networks we use.
-- Added a discussion on the relationship between ResNets and RevNet-type networks, providing evidence that they are closely related.
-- Added additional references from the literature providing evidence of false excessive invariance in non-bijective architectures.
-- Added a random batch of metameric samples to the appendix, to showcase the consistency of our results.

Please let us know if you have any more questions or if there is anything else we can do to make you reconsider your rating.

Thank you once again for your effort.

---

### Public Comment · (anonymous) · 2019-01-19
**Why no CIFAR-10 experiment**

The big jump from MNIST then Imagnet is intriguing.
I tried training with CIFAR-10 using author's open source i-RevNet code, modified the original code to do pooling with DCT II,

and the result is, accuracy is so bad, because everything must fit in 10-class logits.

Imagenet has 1000-class where the logits is much informative.

The author should clarify the limitation of DCT as invertible pooling

---

> ### Author Response · Authors · 2019-01-19
> **Not related to our submission**
>
> 1) The codebase you are referring to is not related to this paper.
>
> 2) The paper presented here makes no claims about Cifar10. Our focus is on Imagenet, which is a much more challenging problem. MNIST is used to illustrate our proposed solution clearly.
>
> 3) The Imagenet model we trained works just as well without invertible DCT pooling. We simply found DCT pooling to make our fi-RevNet conceptually closer to standard ResNets that apply a final global average pooling step, but this is a matter of taste.
>
> 4) iRevNets differ from fiRevNets, we describe this in the paper, so you should not be surprised if they perform differently as well.
>
> It is unfortunate that DCT pooling does not work for you on Cifar10, but neither did we make any claims about it, nor did we experiment with it or provide any implementation of DCT-pooled fiRevNets on Cifar10. However, feel free to drop me an email and I'll try to do what I can to help you.
>
> - Jörn

---

> > ### Public Comment · (anonymous) · 2019-01-20
> > **isn't fiRevNet is iRevNet with average pooling swapped with DCT-II as (spectral) invertible pooling ?**
> >
> > How much does iRevNet differ from fiRevNet?
> > From my understanding, the logits (N-classes output) from DCT-II output is optimized with class labels, and the remaining output is optionally optimized with the proposed loss function.
> > Sorry if I missed something.

---

### Public Comment · ~Ryota_Tomioka1 · 2019-04-30
**Theorem 6**

There might be an error in the proof of Theorem 6. Could the authors clarify what "the properties of conditional mutual information under independence I(y; z_n) =0" means?

To me, it looks like (I might be wrong) the authors are saying I(z_s; y |z_n) = I(z_s; y) if I(y; z_n) = 0 (independence). But I think this is wrong as in the following example:

P(z_n = 0) = P(z_n = 1) = 1/2
P(z_s = 0) = P(z_s = 1) = 1/2

y = z_s if z_n = 1 else 1 - z_s

Then
I(z_s; y) = I(z_n; y) = 0
but
I(z_s; y| z_n) = 1

---

> ### Author Response · Authors · 2019-05-02
> **Response**
>
> Dear Ryota,
>
> Thank you very much for raising this point.
>
> You are right, we have made a mistake in the proof. We have fixed this in the latest revision. Further, we have refined the definition of the adversarial distribution shift to exclude synergetic effects in the interaction information I(y;z_s;z_n). This assumption is also in line with our shiftMNIST experiments where the interaction information between newly introduced features, original digits and labels is >= 0.
>
> We have acknowledged you for pointing this out!
>
> Best,
> Jörn

---

### Public Comment · (anonymous) · 2020-03-07
**what is the difference between the attack in the paper and targeted-adversarial attacks?**

This is a novel method to generate adversarial samples.
However, I could also generate targeted-adversarial attacks such that:
different inputs to a model, but the same output from the model

I just tried targeted-adversarial attacks on vgg-11 trained on ImageNet
x1:  an image of an ant (224x224),  logits is z1=vgg11(x1)
x2:  an image of a bee,   logits is z2=vgg11(x2)
z1 and z2 are very different
Then, targeted PGD-attack is applied to x2
x3 =   x2  + noise, z3=vgg11(x3)
the loss function of the attack is   sum(||z3- z1||^2)
after many..many iterations of PGD, the loss decreases from ~3000 to 0.06399869173765182
x3 still looks the same as x2,  L2_norm(x3-x2) is 14.2280
Now, z3 ~= z1 and predicted class labels are the same, but x3 is a bee and x1 is an ant
different inputs, the same output

---

### Meta-Review · Area_Chair1 · 2018-12-16
**An interesting angle with some issues in terms of execution**

**Confidence:** 4
**Recommendation:** Accept (Poster)

**Metareview:**

This paper studies the roots of the existence of adversarial perspective from a new perspective. This perspective is quite interesting and thought-provoking. However, some of the contributions rely on fairly restrictive assumptions and/or are not properly evaluated.

Still, overall, this paper should be a valuable addition to the program.